# The Effects of Climate Smart Agriculture and Climate Change Adaptation on the Technical Efficiency of Rice Farming—An Empirical Study in the Mekong Delta of Vietnam

**Thanh Tam Ho [1],\* and Koji Shimada [2]** 

[1]  Graduate School of Economics, Ritsumeikan University, 1-1-1 Noji-Higashi, Kusatsu, Shiga 525-8577, Japan
[2]  Faculty of Economics, Ritsumeikan University, 1-1-1 Noji-Higashi, Kusatsu, Shiga 525-8577, Japan; shimada@ec.ritsumei.ac.jp
\*  Correspondence: gr0335xp@ed.ritsumei.ac.jp; Tel.: +81-70-4519-8815

**Abstract:** This study employed the propensity score matching approach to empirically assess the effects of climate smart agriculture participation and climate change adaptation response on the technical efficiency of rice production. Observational data were collected from in-depth interviews with 352 rice farm households in the Mekong Delta, Vietnam. The findings indicate that 71% of local farmers adapted their rice farming to climate change related to salinity intrusion and drought, while 29% of farmers did not. Additionally, only twenty-two rice farmers were typically chosen as participants in the climate smart agriculture pilot program by local government and institutions. The choices for adaptation response and climate smart agriculture participation are significantly influenced by agricultural extension services, belief in climate change, the area of farming land, as well as geographical locations (e.g., province and access to water sources). The results also reveal that climate change adaptation response, including climate smart agriculture participation, played a crucial role in improving technical efficiency of rice production by 13%–14% compared to no adaptation response. Regarding the individual effect of climate smart agriculture participation, participants could achieve higher technical efficiency by 5%–8% compared to non-participants.

**Keywords:** adaptation; climate change; climate smart agriculture; propensity score matching; rice farming; technical efficiency

## 1. Introduction

Rice is the most important crop in the agriculture of Vietnam. It occupies more than 89% of total grain food production. In 2016, Vietnam was one of five major rice exporters, exporting 4.88 million tons, accounting for 11.3% of the total world rice trade. Particularly, Vietnam's Mekong Delta, which encompasses the country's biggest rice producing regions and contributes up to more than 50% of the total rice production of the country [1], is described as one of the most vulnerable regions to climate change in the world. In addition, rice production is the primary livelihood of approximately 60% of the Mekong Delta's residents [2].

However, agriculture is also a major part of the problem related to global warming and climate change. It generates about 19%–29% of total greenhouse gas (GHG) emissions [3]. On the other hand, agriculture is defined as a sector that heavily depends on natural resources and climate, and thus may suffer the adverse impacts of climate change. Specifically, climate change related to increases in temperature, precipitation variation, and the frequency and intensity of extreme weather events are putting pressures on the agricultural system [4]. According to the Intergovernmental Panel on Climate

Change (IPCC) [5], climate change accounts for a significant reduction in renewable surface water and ground water storage in most dry regions. Climate variability in relation to water scarcity results in serious environmental and social consequences that not only threaten agricultural production, but also destabilize rural livelihoods in most regions in the Mekong Delta.

To cope with these risks, farmers decided to perform their adaptation response by adjusting or adapting their farming practices. More specifically, adaptation is pressing to reduce the vulnerability to the adverse impacts of climate change, maintain the rural livelihood of poor communities, and ensure food security [6]. It is reported that farmers taking actions of adaptation could suffer less climate change impacts than those taking no actions [7]. Many case studies using a propensity score matching approach reported that adaptation practices contributed significant benefits in improving crop productivity, income, and food security, as well as reducing the poverty level [8–10].

Furthermore, to adapt or enhance the resilience of the agriculture, reduce greenhouse gas (GHG) emissions, and manage input resources for a sustainable agriculture under the challenges of climate change, climate smart agriculture, which integrates the three dimensions of sustainable development (social, environmental, and economic), is considered as an important approach to achieve food security and agricultural development goals, particularly in developing countries. Yahaya et al. [11] reported that the participation in sustainable agricultural intensification practices played a crucial role in reducing food insecurity status in Northwestern Ghana.

Despite the importance of rice production in the national economy, there has been little study on the effects of adaptation practices, as well as the participation in climate smart agriculture (CSA), in rice farming in Vietnam. Furthermore, competitiveness improvement in rice production is important for maintaining the volume and value of rice exports in the world trade as well as its contributions to a long-term economic development. This will require improvements in technical efficiency level, which is defined as a principle component of competitiveness [12]. Moreover, rice production in the Mekong delta is coping with risks of climatic variability and this situation has been moderated in the case of implementing CSA as well as adaptation practices. Given the importance of CSA and adaptation response in sustaining incomes on rice farming, together with the necessary competitiveness improvements, it is necessary to consider the associated effects on technical efficiency in this sector. This study focuses on determining the effects of climate change adaptation response and the CSA program on the technical efficiency of rice farming at farm level in the Vietnamese Mekong Delta.

## 2. Climate Change Adaptation Response and Climate Smart Agriculture

The IPCC [13] defined adaptation as a change in natural or human systems to respond to actual or expected climate risk and its impacts, as well as to minimize the potential damages. To cope with climate change and its adverse impacts, especially on the agricultural sector, adaptation response is necessary for enhancing the resilience of agricultural production and sustaining rural livelihood [6,7]. The implementation of adaptation practices can not only help rural farmers minimize potential damages in crop yield but also sustain their income and food security status. In details, the adoption of improved maize in Eastern Zambia contributes to significant gains in income (by 78,900 ZMK per ha), consumption expenditure (by 324,690 ZMK per capita), and food security (by 2%) as well as significant reduction in poverty (by 21%) [8]. Adaptation practices in four main provinces of Pakistan had substantial effects on improving net income by 1658–2610 Pakistani rupee per month and increasing wheat yield by 42–65 kg per hectare [9]. Moreover, adaptation practices taken by Pakistani farmers had positively significant effects on food security level (8%–13%) and negatively significant effects on poverty level (3%–6%) [10]. In Vietnamese rice production, Soc Trang farmers who took action on adaptation response to salinity intrusion had a higher annual income, by 34 million Vietnamese Dong, than those who did not [14]. In this study, local adaptation responses including crop improvement practices (e.g., changing sowing or harvesting date, reducing the number of crop plantings, changing fertilizer and chemical use, changing rice varieties, diversifying crop), water management practices,

income diversification practices, and conservation practices have been implemented by rice farmers to cope with climate change related to salinity intrusion and drought in the Mekong Delta.

CSA was introduced by the Food and Agricultural Organization (FAO) in 2010, as an innovation to cope with the challenges of food security and climate change. It aims at increasing the efficiency of resource uses, productivity of the agricultural production system, and enhancing adaptation and resilience, reducing GHG emissions to mitigate climate change for a sustainable agriculture and development. To increase soil productivity, crop yields, and farmers' income in the agricultural sector of Northwestern Ghana, sustainable agricultural intensification practices training was introduced, which can reduce the household food insecurity access score by 11% [11]. Furthermore, the adoption of sustainable agricultural practices in maize farming, involving the promotion of a conservation agriculture package (e.g., crop residue retention, maize-legume rotation) as well as improved crop varieties, played a crucial role in raising yield and income of small-hold farmers in rural Zambia [15]. In this study, the CSA program which aims at producing more with less, reducing GHG emissions, and enhancing adaptation in response to climate change for a sustainable agriculture, was introduced to rice farmers by local government and institutions (e.g., Ministry of Agriculture and Rural Development, Department of Agriculture and Rural Development at provincial level) and agricultural materials companies (e.g., fertilizer companies, chemical companies). For instance, the CSA pilot program associated with reducing seeds as well as fertilizer and chemical uses in rice farming from Binh Dien Fertilizer Joint Stock Company, cooperating with the Ministry of Agriculture and Rural Development, was introduced at thirteen provinces of the Mekong Delta since 2016, with a limited number of participants (only five rice farmers at each province). Furthermore, some agricultural companies and the Department of Agricultural and Rural Development introduced other CSA pilot programs related to integrated pest management (IPM) and change of rice varieties. These CSA pilot programs organized by local governments supported agricultural techniques and knowledge on using rice varieties, fertilizer, and chemicals, as well as IPM to typical farmers through an intensive participatory process considering socio-economic characteristics. In terms of assigning participants, several local rice farmers (introduced by local officers based on the official list of rice farm household in specific communes) were invited to an agricultural extension service training organized by local government and institutions. Then, they were introduced, in detail, to the CSA pilot program, and were asked to voluntarily join. This means that administrators made decisions to recruit local farmers to an introductory training of the CSA pilot program, while farmers made the decision to participate in the program or not based on their intentions.

## 3. Study Site and Data Collection

### 3.1. Study Site

The study was carried out in the Mekong Delta—the main agricultural region of Vietnam. This region has been reported as significantly vulnerable to climate change [16]. The Mekong Delta has a flat terrain, with the average height of 0.7 to 1.2 m. Topography along the Cambodia border is the highest at 2.0 to 4.0 m above sea level. The lower of the central plains is about 1.0 to 1.5 m in elevation. Additionally, there is only 0.3 to 0.7 m elevation in the tidal and coastal areas. Due to this low topography, separated by lots of irrigation canals and being contiguous to the East sea, salinity intrusion and water shortage during the dry season are becoming more serious and are directly affecting rice farming in the delta. Thirteen provinces of the delta have been categorized into four groups of high, moderate, low, and lowest levels of vulnerability to climate change. The lowest vulnerability level in An Giang and Dong Thap provinces are not subjective to projected sea level rise [17]. Therefore, the study aimed at choosing from the rest of three groups. Long An, Ben Tre, and Tra Vinh province were randomly chosen from each of three groups of low, moderate, and high levels, respectively. Ben Tre province has moderate vulnerability to climate change and is located near the coastal region, where water sources are often intruded upon by saline water, while Long An

province has low vulnerability to climate change and is located further inland, where water sources are only slightly affected by salinity intrusion. Tra Vinh province has high vulnerability to climate change and is located at the coastal region, where salinity intrusion becomes serious. In each province, two districts were randomly selected (Figure 1). Then two communes were continuously picked up from each district.

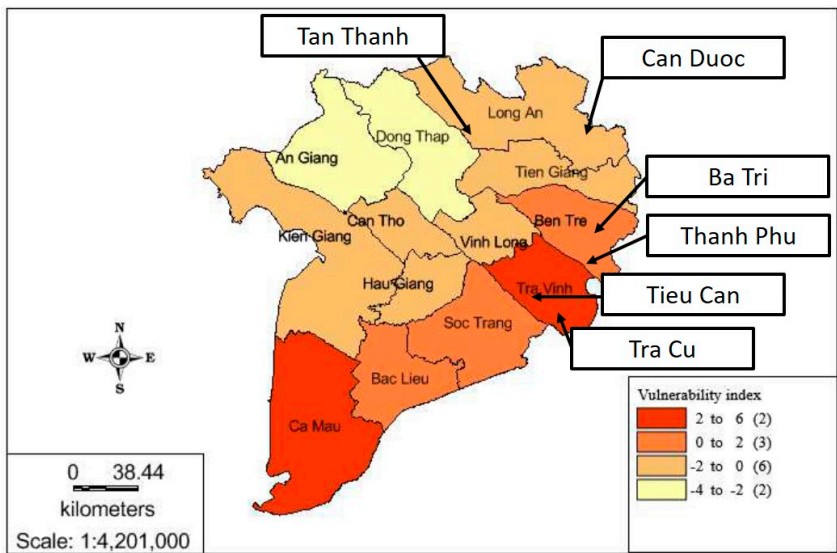

**Figure 1.** Map of the study areas. Source: Thuy and Anh [17].

*3.2. Data Collection*

A field survey was carried out in February 2018 in three provinces of the Mekong Delta, more specifically Long An, Ben Tre, and Tra Vinh. The cross-sectional data of 361 households via face-to-face interviews with the structured questionnaire were selected. The respondents were selected based on the guidance of village leaders from the official household lists of each commune. These interviewed households have different access to water sources. It was divided into three levels of near, medium, and far access to water sources based on the distance to main, secondary, and small irrigation systems, respectively. In detail, rice farms located near main rivers or main irrigation systems were defined as having near access to water sources. Rice farms located far from main rivers or main irrigation systems, but relatively close to secondary irrigation canals were classified as having medium access to water source, while others located far from the main and secondary irrigation systems were classified as having far access to water source. Additionally, only one individual from a household (mainly head of household) was surveyed. The structured questionnaire included four sections: household characteristics, climate change awareness, climate change adaptation behavior, and agricultural production. We dropped nine observations due to incomplete information. The final sample of 352 rice farm households was used for data analysis.

## 4. Methodological Framework and Data Description

*4.1. Methodological Framework*

Both interviews and CSA pilot program in this study were assigned by administrator selection with lack of randomization. This could face a problem of selection bias ("selection bias or hidden bias is essentially a problem created by the omission in statistical analysis of important variables, and omission renders nonrandom the unobserved heterogeneity reflected by an error term in regression equations." [18] (p. 357). Sources of selection bias are categorized into self-selection, researcher selection, administrative selection, geographic selection, measurement selection, and attrition selection.) that may lead to not only endogeneity bias or confoundedness, but also bias estimation of causal effects.

Therefore, it is important to apply the propensity score matching approach proposed by Rosenbaum and Rubin [19] for adjusting selection bias, as well as balancing data in terms of control covariance before evaluating the treatment effects.

In recent years, there has been increasing interest in estimating the causal effects with the elimination of selection bias in observational studies or studies without randomization using the propensity score matching (PSM) approach [20]. Several studies in social and health sciences employed this approach for assessing the role of kindergarten retention on children's social-emotional development [21], the effects of Alcoholics Anonymous [22], the impact of small school size on mathematics achievement [23], and the impact of teenage alcohol use on education attainment [24]. In the agricultural sector and current environmental issues, the benefits of climate change mitigation and adaptation strategies were also evaluated by the PSM approach. More specifically, Ali and Erenstein [10] applied PSM to estimate the effects of the number of adaptation measures on household food security and poverty in Pakistan. Similarly, the impacts of climate-risk mitigating practices on wheat productivity, income and poverty in the Himalayan region of Pakistan [25], the benefits of adoption associated with improved maize varieties on crop incomes, consumption expenditure, and food security in Eastern Zambia [8], and the impacts of climate change adoption on wheat productivity and net crop income in Pakistan [9] were also investigated by PSM. Furthermore, it was employed to evaluate the impacts of export horticulture farming on calorie intake per capita among small-hold farmers in the Eastern and Central provinces of Kenya [26]. In dairy farming, the role of innovations in farm economic sustainability in Ireland was evaluated by a generalized propensity score [27]. Therefore, this study attempts to evaluate the effects of climate change adaptation response and the CSA program on the technical efficiency of rice farming in the Mekong Delta, Vietnam using a PSM approach.

Conceptually, the process of PSM has three analytic steps [18]. The propensity scores for each subject are estimated by a binary logistic model in the first step. Then matching treated and control subjects who share a relatively similar propensity score and estimating the treatment effects are undertaken in the second step. Post-matching analysis is also examined in the final step.

Firstly, the binary propensity score model was run to investigate rice farmers' preference of CSA participation and climate change adaptation response and to estimate the propensity score as the predicted probability of treatment (see equation (1)). Several previous studies also used the binary logistic model to determine key factors affecting farmers' decisions on adaptation response to climate change [6,28,29] as well as to identify influencing factors on farmers' participation decisions [11,30–32]. The propensity score model is generally described as follows:

$$P(D_i/X_i = x_i) = \frac{e^{\beta i X i}}{1 + e^{\beta i X i}} = \frac{1}{1 + e^{-\beta i X i}} \tag{1}$$

where $D_i$ represents CSA participation or adaptation response to climate change with respect to salinity intrusion and drought, with 1 denoting adaptation or CSA (treated group) and 0 denoting non-adaptation or non-CSA (control group). $X_i$ is a vector of covariates for each subject. The selection of conditioning covariate in the propensity model might affect both the balance of the propensity score between the control and treated group, and the final estimate of the treatment effects. Therefore, efforts to consider all substantively relevant factors and the best conditioning variables in the model should be highlighted.

After the propensity scores are predicted by the propensity score model, the scores are used to match treated and control subjects and then the treatment effects are estimated in the second step. More specifically, a pair of treated and control subjects sharing the same propensity scores are essential considered as comparable for matching, even though they still could differ on specific observed covariates. Among various matching algorithms, nearest neighbor matching operates as one-to-one pair matching. A control subject *j*, from the set of control subjects $I_0$, is matched with one nearest treatment subject *i* from the set of treatment subjects $I_1$, when the single absolute difference of propensity scores among possible pairs of control and treatment subject is the smallest. Once a

control subject *j* is found to match to treatment subject *i*, *j* is deleted from $I_0$ without replacement. A neighborhood $C(P_i)$ is defined to be:

$$C(P_i) = min\ (P_i - P_j) \qquad (2)$$

The effect of treatment for each observation can be expressed as follows:

$$Y_i\ (1) - Y_i\ (0) \qquad (3)$$

where, $Y_i$ is defined as the observed outcome for the subject *i*, especially $Y_i\ (1)$ is the observed outcome for the treated group and $Y_i\ (0)$ is the observed outcome for the control group.

The average treatment effect (ATE) or average causal effect, which is defined as the average effect at the population level [33], can be written as:

$$E\ [Y_i\ (1) - Y_i\ (0)] \qquad (4)$$

The average treatment effect for the treated (ATT) is estimated as the average effect of treatment on those subjects who received the treatment [33]. Therefore, it is critical for investigating the benefit of a policy or intervention. In this study, the ATT was specified as the average difference in the outcome (i.e., technical efficiency) of rice farmers who performed CSA participation or adaptation response and rice farmers who did not. The ATT can be written as:

$$E\ [Y_i\ (1) - Y_i\ (0)/D_i = 1] \qquad (5)$$

Furthermore, the kernel-based matching involving kernel matching and local linear matching deploys information from all possible control subjects as comparators in the matching procedure for estimating the ATT [18], as compared to nearest neighbor matching. Specifically, the kernel matching algorithm employs a kernel estimator with complicated functions developed from non-parametric regression methods [34,35] for capturing the weighted average for a focal point [18]. This matching algorithm operates as one-to-many matching by estimating the weighted average of the outcome for all control subjects and then comparing that weighted average of the outcome for the treated subjects. The difference between two terms provides an estimate of ATT, which can be specified as follows:

$$ATT = (1/n) \sum [Y_i\ (1) - \sum W\ (i, j)\ Y_i\ (0)] \qquad (6)$$

where $W\ (i, j)$, which is defined as the weight or the distance on the estimated propensity scores between the control and treated group, are determined by the kernel estimator.

The nearest neighbor matching and kernel matching are considered as the two main matching methods in this study.

For checking the robustness and adequacy of the results, sensitivity analysis and quality of matching checks are employed in the final step. With regards to sensitivity analysis, it is important to ask whether the unobserved covariate would markedly alter the robustness of the results or their sensitivity to selection bias [36]. Theoretically, a variety of sensitivity analysis methods comprising McNemar's test, Wilcoxon's signed rank test, and the Hodges–Lehmann point and interval estimates were developed by Rosenbaum [18]. Sensitivity analysis proposes to determine several possible values of gamma $\Gamma$ (a measure of a degree to be free of selection bias) and consider how the conclusions of the study might change at a given significant level. As explained by Rosenbaum [37], a study is sensitive if $\Gamma$ values close to 1 would alter the inferences. Meanwhile, a study is insensitive if extreme $\Gamma$ values are needed to change the inferences. Moreover, the *p*-value of the Wilcoxon's signed rank test illustrates how robust, to a plausible range of selection bias, the estimated treatment effects are. If the *p*-value is less than the given significant level, we can reject the null hypothesis of no treatment effect.

For checking the quality of matching, the balancing test is required. Precisely, the test was first mentioned in Rosenbaum and Rubin [38] to check the balance in propensity scores between the control and treated group. The aim of the balancing test is to check if the propensity score is an adequate balancing score, or if the study properly uses a corrective propensity score model and matching algorithm. Among the various common balancing tests, the standardized test of differences is specified to confirm the balance between the treated and control groups with the following formula:

$$B_{before}(X) = \frac{\overline{X}_T - \overline{X}_C}{\sqrt{\frac{[V_T(X) + V_C(X)]}{2}}} \cdot 100 \tag{7}$$

$$B_{after}(X) = \frac{\overline{X}_{TM} - \overline{X}_{CM}}{\sqrt{\frac{[V_T(X) + V_C(X)]}{2}}} \cdot 100 \tag{8}$$

where $\overline{X}_T$ and $\overline{X}_C$ are the means for the unmatched treated and control groups, respectively; $\overline{X}_{TM}$ and $\overline{X}_{CM}$ are the means for the matched treated and control groups, respectively; and $V_T(X)$ and $V_C(X)$ are the corresponding variances of treated and control groups, respectively. $B_{before}(X)$ and $B_{after}(X)$ represent the percentage of the mean standardized difference or bias between the treated and control groups before and after matching, respectively. Specifically, the standardized bias is defined as the difference in means of a covariate $X_i$ between the treated and control subject. Then the matching quality can be evaluated by a reduction in the standardized bias. A significant reduction in bias implies that the propensity score is an adequate balancing score. Meanwhile, if the bias remains, either the propensity score models should be estimated using a different method, or an alternative matching algorithm should be used, or both. All estimations were compiled by STATA version 15.0 (StataCorp LLC, College Station, TX, USA).

### 4.2. Data Description

The explanatory variables in the binary logistic model were selected based on literature on the performance of climate change adaptation response and CSA participation in agricultural production. These variables included education level, farming experience, agricultural extension services, belief in climate change, trust in public adaptation, social norm, and geographical settings (e.g., province and access to water source). More educated farmers were reported to affect the choice of climate change adaptation [29,39]. They also suggested that the preference of adaptation choice was positively influenced by access to information (e.g., agricultural extension services). Additionally, belief of climate change was reported to pose a significant influence on adaptation behavior [28,40,41]. Ho and Shimada [15] found that the higher the pressures local farmers heard or saw from their neighbors, relatives or friends, the less they adapted their rice farming in response to climate change. Regarding farm characteristics, such as geographical settings, some studies have identified that farmers living in different locations generally applied different adaptation practices [39,42]. Moreover, land size was shown to have a negative influence on the performance of adaptation [43] or a positive one affecting the adoption of sustainable agricultural intensification practices [8].

In terms of explanatory variables in the binary logistic models (Table 1), some demographic variables related to education level and farming experience were expected to influence the choice of CSA participation and adaptation response to climate change. The performances were also expected to be positively affected by access to formal information from agricultural extension services. In addition, cognitive factors, such as belief in climate change, were assumed to positively affect the choice of CSA participation and adaptation response. Additionally, trust in public adaptation and social norm were hypothesized to be positively significant in the choice of adaptation response. Regarding farm characteristics, farmers living in different geographical locations at the provincial level and micro-level (e.g., access to water source) were expected to be different in their choice of adaptation response and

CSA participation. Furthermore, the area of farming land was assumed to positively affect the choice of CSA participation and adaptation response.

**Table 1.** Description of explanatory variables in the binary logistic models.

| Variables | Description |
|---|---|
| Adaptation | Dependent variable—dummy variable, 1 represents farmer performs adaptation, 0 represents farmer performs no adaptation |
| Climate smart agriculture (CSA) program | Dependent variable—dummy variable, 1 represents farmer participates in CSA pilot program, 0 represents farmer does not participate in CSA pilot program |
| Education | Number of years in school |
| Experience | Number of years of rice farming experience |
| Extension | Dummy variable, 1 represents farmers participate in agricultural extension services, 0 represents otherwise |
| Belief in climate change | Five-point Likert scales measure, from 1, meaning strongly disagree, to 5, meaning strongly agree with the adverse impacts of climate change on rural livelihood |
| Trust in public adaptation | Five-point Likert scales measure, from 1, meaning strongly disagree, to 5, meaning strongly agree with the effectiveness of public adaptation |
| Social norm | Five-point Likert scales measure, from 1, meaning strongly disagree, to 5, meaning strongly agree with the dependence of performing or not performing adaptation practices upon friends, relatives, and neighbors |
| Farm area | Total area of rice land, measures in hectare |
| Access to water sources (Near) [a] | Distance to water source intuitively evaluated by rice farmer—dummy variable, 1 represents farm locates in near, 0 represents otherwise |
| Access to water sources (Medium) [a] | Distance to water source intuitively estimated by rice farmer—dummy variable, 1 represents farm locates in medium, 0 represents otherwise |
| Region (Long An) [b] | Farm location—dummy variable, 1 represents farmer locate in Long An province, 0 represents otherwise |
| Region (Ben Tre) [b] | Farm location—dummy variable, 1 represents farmer locate in Ben Tre province, 0 represents otherwise |

Note: [a] Access to water sources (Far) represents a reference or comparison category. [b] Region (Tra Vinh) represents a reference or comparison category.

To simply estimate the differences between the control and treatment groups, the difference test in means (*t*-statistics) was used.

Regarding the treatment variable of adaptation response, the *p*-value of *t*-statistics from Table 2 showed that farmer characteristics including experience, extension, social norm, and trust in public adaptation, and farm characteristics, such as the area of rice farming and geographical location at the provincial level significantly, differed between farmers who performed adaptation response and those who did not. More specifically, adapting farmers were more likely to have less farming experience than non-adapting farmers. This can be explained that younger farmers with less experience are presumably more willing to adapt innovations than experienced farmers. Furthermore, rice farmers in Ben Tre province were less likely to perform adaptation compared to other provinces. On the other hand, more farmers who performed adaptation response preferred to participate in agricultural extension services than farmers who did not. In addition, the trust in public adaptation and social norm within adapting farmers were higher than those within non-adapting farmers. The area of farming land owned by adapting farmers was larger than that owned by non-adapting farmers.

Regarding the treatment variable of CSA participation, the *p*-value of *t*-statistics from Table 3 displays that farmer characteristics, including education, extension, belief in climate change, and farm characteristics, such as the area of rice farming and geographical location at the provincial level and micro-level, were significantly different between farmers who participated in the CSA pilot program and those who did not. More specifically, participating farmers were more likely to have a higher level of education than non-participating farmers. This can be explained by a tendency that more educated farmers were more willing to participate in the pilot CSA program. Furthermore, CSA participants preferred to join in agricultural extension services than CSA non-participants. In addition, belief in climate change within CSA participating farmers was higher than that within non-participating farmers. The area of farming land owned by CSA participants was larger than that owned by non-participants. Rice farms of CSA participants were generally located at nearer distance to water sources than those of non-participants. On the other hand, the rice farms of CSA non-participants were more often located

at medium distance to water sources than those of participants. Furthermore, rice farmers in Ben Tre province were less likely to participate in the CSA pilot program compared to other provinces.

**Table 2.** Difference between control and treatment group regarding adaptation response.

| Variables | Non-Adapting (*n* = 103) | Adapting (*n* = 249) | Difference (0 vs. 1) | *p*-Value | | |
|---|---|---|---|---|---|---|
| | Mean | Mean | | H: D [a] < 0 | H: D ≠ 0 | H: D > 0 |
| Education | 5.6311 | 6.0080 | −0.3770 (0.4068) | 0.1774 | 0.3547 | 0.8226 |
| Experience | 28.9806 | 26.1407 | 2.8400 ** (1.2823) | 0.9863 | 0.0274 | 0.0137 |
| Extension | 0.3786 | 0.6305 | −0.2519 *** (0.0568) | 0.0000 | 0.0000 | 1.0000 |
| Belief in climate change | 3.6699 | 3.7871 | −0.1172 (0.1145) | 0.1533 | 0.3067 | 0.8467 |
| Social norm | 2.6019 | 2.7871 | −0.1852 * (0.1148) | 0.0539 | 0.1077 | 0.9461 |
| Trust in public adaptation | 3.5437 | 3.8072 | −0.2635 ** (0.1237) | 0.0169 | 0.0338 | 0.9831 |
| Farm area | 1.0466 | 1.3607 | −0.3141 ** (0.1610) | 0.0259 | 0.0519 | 0.9741 |
| Access to water sources (Near) | 0.4563 | 0.4538 | 0.0025 (0.0585) | 0.5170 | 0.9660 | 0.4830 |
| Access to water sources (Medium) | 0.4078 | 0.4900 | −0.0822 (0.0584) | 0.0802 | 0.1605 | 0.9198 |
| Region (Long An) | 0.3495 | 0.3293 | 0.0202 (0.0535) | 0.6420 | 0.7159 | 0.3580 |
| Region (Ben Tre) | 0.4078 | 0.2972 | 0.1106 ** (0.0549) | 0.9776 | 0.0448 | 0.0224 |

Note: Standard errors are in parentheses. *, **, and *** means significance with confidence interval at 90%, 95%, and 99%, respectively. [a] D denotes difference (0 vs. 1).

**Table 3.** Difference between control and treatment group regarding CSA participation.

| Variables | Non-CSA (*n* = 330) | CSA (*n* = 22) | Difference (0 vs. 1) | *p*-value | | |
|---|---|---|---|---|---|---|
| | Mean | Mean | | H: D [a] < 0 | H: D ≠ 0 | H: D > 0 |
| Education | 5.8333 | 6.8636 | −1.0303 * (0.7635) | 0.0890 | 0.1781 | 0.9110 |
| Experience | 26.8849 | 28.2727 | 1.3879 (2.4259) | 0.2838 | 0.5676 | 0.7162 |
| Extension | 0.5303 | 0.9545 | −0.4242 *** (0.1073) | 0.0000 | 0.0000 | 1.0000 |
| Belief in climate change | 3.7121 | 4.3636 | −0.6515 *** (0.2128) | 0.0012 | 0.0024 | 0.9988 |
| Farm area | 1.2025 | 2.2636 | −1.10612 *** (0.2989) | 0.0002 | 0.0004 | 0.9998 |
| Access to water sources (Near) | 0.4424 | 0.6364 | −0.1939 ** (0.1095) | 0.0387 | 0.0773 | 0.9613 |
| Access to water sources (Medium) | 0.4758 | 0.3182 | 0.1576 * (0.1098) | 0.9239 | 0.1523 | 0.0761 |
| Region (Long An) | 0.3303 | 0.4091 | −0.0788 (0.1042) | 0.2250 | 0.4499 | 0.7750 |
| Region (Ben Tre) | 0.3424 | 0.1364 | 0.2061 ** (0.1032) | 0.9767 | 0.0467 | 0.0233 |

Note: Standard errors are in parentheses. *, **, and *** means significance with confidence interval at 90%, 95%, and 99%, respectively. [a] D denotes difference (0 vs. 1).

Consequently, these differences illustrate that selection bias seemed to exist in the observational data. Hence, the propensity score matching approach was necessary to be deployed to reduce the problem of selection bias in estimating the treatment effects in the study.

## 5. Results and Discussion

### 5.1. Adaptation Response, CSA Participation, and Technical Efficiency of Rice Farming

#### 5.1.1. Adaptation Response and CSA

Regarding the CSA pilot program, there were twenty-two rice farmers who were typically chosen as participants. The CSA pilot programs involve the application of IPM, change of rice variety, and reductions of seed, fertilizer, and chemical use with the aim of improving crop productivity, reducing production cost, and reducing GHGs emissions under climate change. More specifically, there were nine CSA participants in Long An province with the programs of IPM (two participants), producing with less (i.e., reducing seeds, fertilizer, and chemical use) (five participants), and changing rice variety (two participants). Three farm households in Ben Tre province participated in the CSA program of changing rice variety and reducing fertilizer and chemical use. Ten farm households in Tra Vinh province participated in the CSA program with IPM (three participants), producing with less (three participants), and changing rice variety (four participants).

To cope with risks of climate change associated with salinity intrusion and drought in rice farming, local farmers have implemented individual or collective adaptation measures (Figure 2).

Specifically, approximately 71% of farmers typically decided to adapt their rice farming by improving crops (e.g., changing sowing or harvesting date, reducing the number of crop plantings, changing fertilizer and chemical use, changing crop variety, and diversifying crop), changing irrigation schedule, diversifying source of income, changing the area of farming land, and soil conservation. Meanwhile, 29% of the others did not adapt.

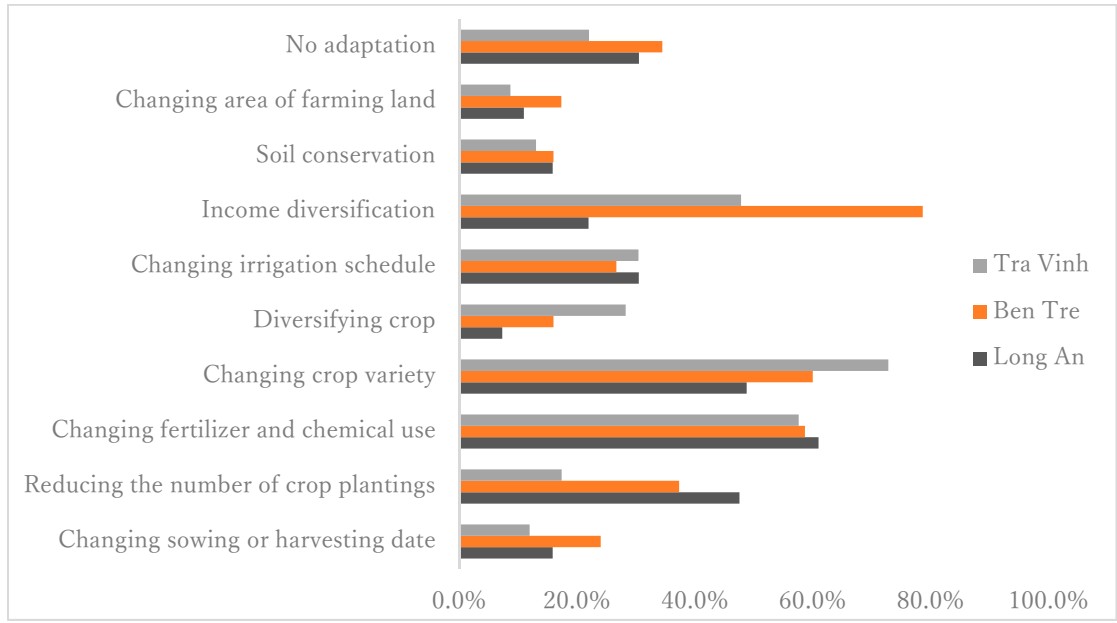

**Figure 2.** Adaptation practices implemented by local rice farmers.

In details, Ben Tre farmers (79%) were more likely to diversify their income sources (i.e., shifting from rice production to livestock, grass cultivating, or shrimp farming in part or in total) than Long An farmers (22%) and Tra Vinh farmers (48%). Meanwhile, Tra Vinh farmers preferred to change rice variety (occupying 73%) (i.e., change traditional rice variety to salt-tolerant rice variety) and diversifying their crops (i.e., farming an integrated rice–shrimp) (occupying 28%). Furthermore, Long An farmers (48%) tended to reduce the number of crop plantings (i.e., cultivating two rice crop per year) compared to Ben Tre farmers (37%) and Tra Vinh farmers (17%).

5.1.2. Technical Efficiency of Rice Farming

Ho and Shimada [44] investigated the technical efficiency of rice farming in the Vietnamese Mekong Delta to be 0.7725 through a parametric approach of stochastic frontier analysis (SFA) (Table 4). More specifically, all four input variables of the amount of seed, amount of fertilizer use, the cost of chemical use, and the cost of irrigation water use were reported to pose positively significant effects on the output level of rice yield. In addition, the estimated return-to-scale, computed as the sum of coefficients from the Cobb–Douglass production frontier model, was 0.3801, implying that rice farms in the Mekong delta were operating at decreasing returns to scale.

**Table 4.** The technical efficiency estimated by the stochastic frontier analysis approach.

| Technical Efficiency | Value |
| --- | --- |
| Mean technical efficiency | 0.7725 |
| Standard deviation | 0.1564 |
| Minimum | 0.1497 |
| Maximum | 0.9543 |

With regards to the treatment variable of adaptation response, the result from Table 5 depicts that the average difference in technical efficiency between adapting and non-adapting farmers is statistically negative at the significant level of 0.01. This implies that farmers who have not performed adaptation response have a 15.29% lower technical efficiency than farmers who have performed adaptation response to climate change.

**Table 5.** Difference test of means in technical efficiency by adaptation response.

| Non-adapting (*n* = 103) | Adapting (*n* = 249) | Difference (0 vs. 1) | *p*-Value | | |
| --- | --- | --- | --- | --- | --- |
| Mean | Mean | | H: D [a] < 0 | H: D ≠ 0 | H: D > 0 |
| 0.6643 | 0.8172 | −0.1529 *** (0.0164) | 0.0000 | 0.0000 | 1.0000 |

Note: Standard errors are in parentheses. *** means significance with confidence interval at 99%. [a] D denotes difference (0 vs. 1).

With regards to the treatment variable of CSA participation, the result from Table 6 shows that the average difference in technical efficiency between CSA and non-CSA farmers was statistically negative at the significant level of 0.01. This means that farmers who did not participate in the CSA pilot program had a 12.07% lower technical efficiency than farmers who did participate in the CSA pilot program.

**Table 6.** Difference test of means in technical efficiency by CSA participation.

| Non-CSA (*n* = 330) | CSA (*n* = 22) | Difference (0 vs. 1) | *p*-Value | | |
| --- | --- | --- | --- | --- | --- |
| Mean | Mean | | H: D [a] < 0 | H: D ≠ 0 | H: D > 0 |
| 0.7649 | 0.8857 | −0.1207 *** (0.0339) | 0.0002 | 0.0004 | 0.9998 |

Note: Standard errors are in parentheses. *** means significance with confidence interval at 99%. [a] D denotes difference (0 vs. 1).

With such results of the difference test in means, the innovations in agricultural production with the CSA pilot program and adaptation practices in response to climate change are shown to have significant effects on technical efficiency of rice farming.

*5.2. The Effects of Adaptation Response and CSA on Technical Efficiency*

5.2.1. The Propensity Score Models

To overcome the problem of selection bias, the PSM approach was used to estimate the effects of adaptation response and CSA participation on technical efficiency of rice farming. Firstly, the binary logistic models were run to predict the propensity score estimates for two treatment variables of adaptation and CSA (Table 7).

According to farmers' characteristics, the variable of extension was positively significant in the propensity score models. This means that farmers who participated in agricultural extension services were more willing to adapt their rice farming in response to climate change, as well as willing to participate in the pilot CSA program. The variable of belief in climate change had a positive significance in model 1. This implies that farmers who participated in the CSA pilot program had higher belief in climate change than farmers who did not. Furthermore, the area of farming land was positively significant. This indicates that the larger the area of farming land farmers owned, the more they were willing to participate in CSA pilot program. Regarding geographical location at a micro-level, the dummy variable of access to water source (near) was positively significant in model 2. This means that farmers who performed their adaptation response were often located a near distance to water sources, as compared to farmers who did not.

**Table 7.** The propensity score models.

| Variables | Model 1 (CSA) | | Model 2 (Adaptation) | |
|---|---|---|---|---|
| | Coefficient | Marginal | Coefficient | Marginal |
| Education | 0.0348 | 0.0018 | −0.0220 | −0.0041 |
| | (0.0707) | (0.0036) | (0.0395) | (0.0074) |
| Experience | 0.0271 | 0.0014 | −0.0186 | −0.0035 |
| | (0.0218) | (0.0011) | (0.0115) | (0.0021) |
| Extension | 2.6239 ** | 0.1322 ** | 0.8160 *** | 0.1523 *** |
| | (1.0450) | (0.0551) | (0.2594) | (0.0461) |
| Belief in climate change | 0.8119 ** | 0.0409 ** | 0.0254 | 0.0047 |
| | (0.3367) | (0.0172) | (0.1292) | (0.0241) |
| Farm area | 0.2659 * | 0.0134 * | 0.1695 | 0.0316 |
| | (0.1397) | (0.0070) | (0.1150) | (0.0213) |
| Region_Long An | −0.3519 | −0.0177 | −0.3981 | −0.0243 |
| | (0.5733) | (0.0289) | (0.3219) | (0.0597) |
| Region_Ben Tre | −0.6540 | −0.0330 | −0.4215 | −0.0787 |
| | (0.7148) | (0.0362) | (0.3257) | (0.0603) |
| Access to water source (Near) | 0.5987 | 0.0302 | 0.6206 | 0.1158 |
| | (1.1394) | (0.0575) | (0.4407) | (0.0813) |
| Access to water source (Medium) | −0.1589 | −0.0080 | 0.9792 ** | 0.1828 ** |
| | (1.1139) | (0.0597) | (0.4439) | (0.0808) |
| Trust in public adaptation | - | - | 0.1581 | 0.0295 |
| | | | (0.1204) | (0.0222) |
| Social norm | - | - | 0.1175 | 0.0219 |
| | | | (0.1290) | (0.0240) |
| Constant | −9.5085 *** | - | −0.5222 | - |
| | (2.2889) | | (0.9206) | |
| Number of observations | 352 | | 352 | |
| Likelihood ratio chi$^2$(9) | 37.70 | | 35.14 | |
| Pseudo $R^2$ | 0.0000 | | 0.0002 | |
| Probability > chi$^2$ | 0.2291 | | 0.0826 | |

Note: Standard errors are in parentheses. *, **, and *** means significance with confidence interval at 90%, 95%, and 99%, respectively.

### 5.2.2. The Average Treatment Effects for the Treated by Propensity Score Matching

The next step of analysis is estimating the treatment effects for the treated (ATT) based on propensity score estimates through two common matching procedures: nearest neighbor matching and kernel matching, as well as checking overlap or the common support.

Regarding the treatment variable of adaptation response, Table 8 presents that the ATT from both matching algorithms are of statistically positive significance. This indicates that adaptation response has a positive effect on the technical efficiency of rice farming. Regarding the nearest neighbor matching, the average ATT of 0.1357 shows that farmers who have responded to climate change effects related to salinity intrusion and drought, by the application of adaptation practices, have a 13.57% higher technical efficiency compared to those who have not responded. On the other hand, the average ATT of 0.1266 from the kernel matching shows that farmers who performed adaptation response can achieve higher technical efficiency by 12.66% compared to those who did not.

The authors acknowledge the limitation in control observations for not providing enough comparator information for matching. The data with less non-adaptation farmers (103 farmers) compared to adaptation farmers (249 farmers) are mainly due to the wide provision of information related to climate change and adaptation strategies from local government and institutions through vertical social networks, as well as agricultural extension services or trainings in the study site.

In terms of CSA participation, Table 9 presents that the ATT from both matching algorithms are statistically positively significant. It reveals that CSA is significantly effective in improving the technical efficiency level of rice production. Regarding the nearest neighbor matching, the average ATT of 0.0451 indicates that farmers who participated in the pilot CSA program, organized by local government and institutions as well as private companies, have a 4.51% higher technical efficiency compared to those who did not. On the other hand, the average ATT of 0.0820 from the kernel matching

shows that CSA-participating farmers can achieve a 8.2% higher technical efficiency compared to non-CSA participating farmers. A possible reason why the CSA program has a relatively low effect on promoting technical efficiency is that farmers in the CSA program only employed an application of IPM, changed rice variety, and reduced seed, fertilizer and chemical use, while non-CSA farmers may have performed a variety of climate change adaptation responses, including crop improvement (i.e., changed fertilizer and chemical use, changed rice variety, changed sowing or harvesting date, reduced the number of crop plantings, diversified crops), water management, income diversification, and soil conservation to cope with salinity and drought.

**Table 8.** Treatment effects of adaptation response on technical efficiency.

| CSA Participation | Nearest Neighbor Matching | Kernel Matching |
|---|---|---|
| ATE | 0.1419 *** | 0.1299 |
| CSA (1 vs 0) | (0.0193) | (NC) |
| ATT | 0.1357 *** | 0.1266 ** |
| CSA (1 vs 0) | (0.0234) | (NC) |

Note: ** and *** means significance with confidence interval at 95% and 99%, respectively. NC means not comparable due to bootstrap. ATE means Average treatment effect while ATT means Average treatment effect for the treated.

**Table 9.** Treatment effects of CSA participation on technical efficiency.

| CSA Participation | Nearest Neighbor Matching | Kernel Matching |
|---|---|---|
| ATE | 0.1317 *** | 0.0841 |
| CSA (1 vs 0) | (0.0187) | (NC) |
| ATT | 0.0451 * | 0.0820 ** |
| CSA (1 vs 0) | (0.0250) | (NC) |

Note: *, **, and *** means significance with confidence interval at 90%, 95%, and 99%, respectively. NC means not comparable due to bootstrap.

Taken together, Figures 3 and 4 show that the distributions of the propensity scores of two treatment variables drop in the common support region. This implies that there is a considerable overlap in the distribution of propensity scores between the treated and control groups in two treatment variables.

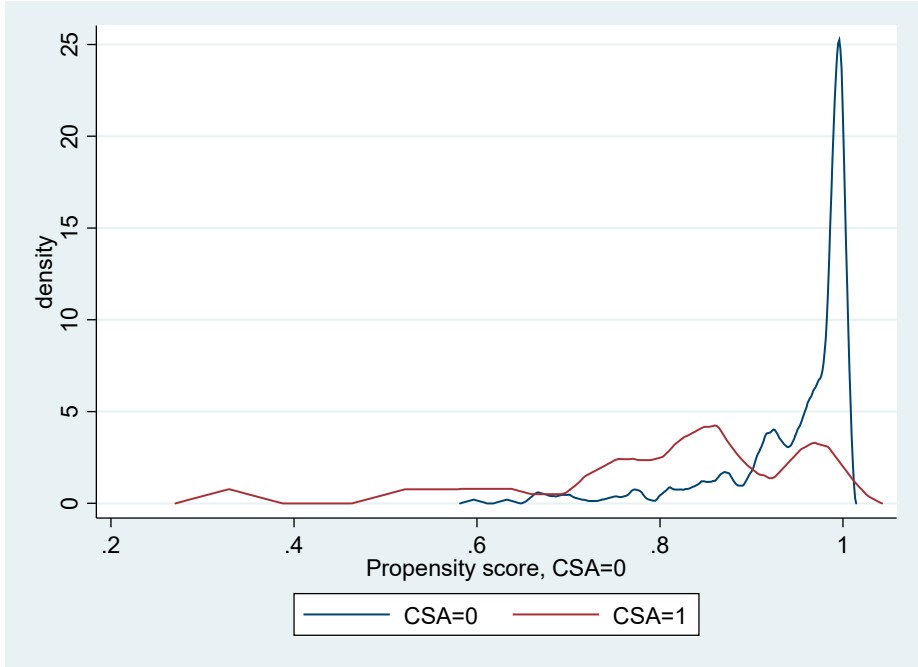

**Figure 3.** Distribution of propensity scores of CSA.

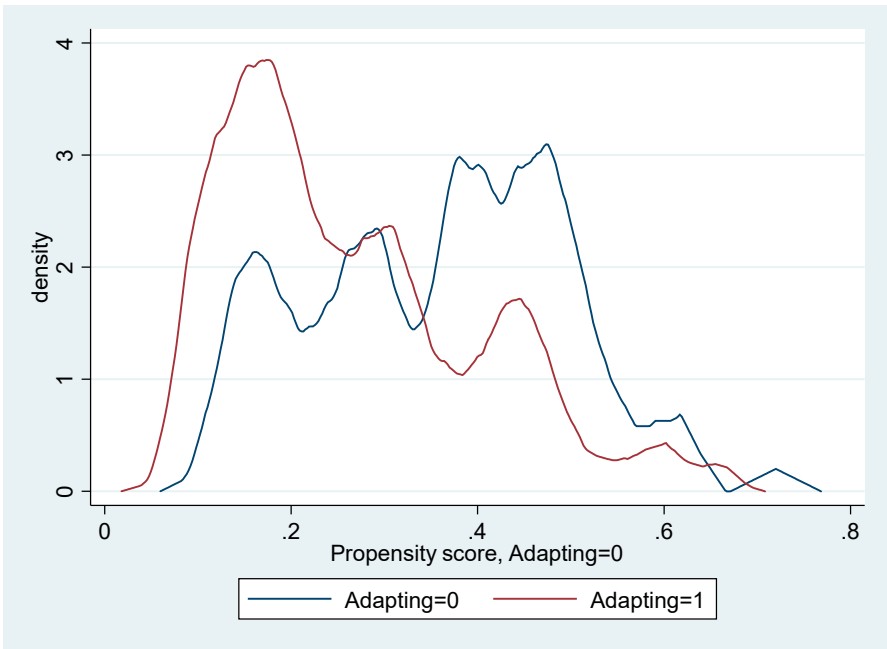

**Figure 4.** Distribution of propensity scores of adaptation response.

### 5.2.3. Sensitivity Analysis and Balancing Test

Finally, checking of the selection bias and the quality of the matching algorithm, overall, of this study was conducted. With regard to sensitivity analysis, several possible values of gamma $\Gamma$ (log odds of differential assignment due to unobserved factors), the upper and lower bounds of significant level, or the *p*-value of Wilcoxon's signed rank test (sig+ and sig−), the upper and lower bounds of the Hodges–Lehmann point estimate (t-hat+ and t-hat−), and the upper and lower bounds of the 95% confidence interval of the Hodge–Lehmann interval estimate (CI+ and CI−) are summarized in Table 10. As presented, the significance interval becomes uninformative at the value of $\Gamma = 6$. Because of a large value of $\Gamma = 6$, it can be reported that the study is robust against selection bias. Moreover, the *p*-value (sig+ and sig−) at $\Gamma = 6$ is lower than the significant level of 0.05, the null hypothesis of no treatment effect can be; therefore, rejected. This indicates that even a substantial amount of unobserved covariate would not change the estimated treatment effects.

**Table 10.** Result of the Rosenbaum sensitivity analysis.

| Gamma | sig+ | sig− | t-hat+ | t-hat− | CI+ | CI− |
|---|---|---|---|---|---|---|
| 1 | 0 | 0 | 0.1219 | 0.1219 | 0.1028 | 0.1411 |
| 2 | $2.9 \times 10^{-9}$ | 0 | 0.0786 | 0.1651 | 0.0570 | 0.1855 |
| 3 | 0.000054 | 0 | 0.0532 | 0.1889 | 0.0297 | 0.2108 |
| 4 | 0.0049 | 0 | 0.0363 | 0.2050 | 0.0098 | 0.2277 |
| 5 | 0.0518 | 0 | 0.0228 | 0.2169 | −0.0060 | 0.2403 |
| 6 | 0.1947 | 0 | 0.0115 | 0.2262 | −0.0184 | 0.2505 |

Note: sig+ and sig−: the upper and lower bounds of significant level, or the *p*-value of Wilcoxon's signed rank test; t-hat+ and t-hat−: the upper and lower bounds of the Hodges–Lehmann point estimate; and CI+ and CI−: the upper and lower bounds of the 95% confidence interval of the Hodge–Lehmann interval estimate.

With respect to the balancing test, the test of standardized differences is employed to confirm an obvious reduction in mean standardized difference or bias of each covariate between unmatched (before matching) and matched (after matching) treatment variables of adaptation response (Table 11)

and CSA participation (Table 12). The percentage of reduction in bias of each covariate for two propensity score models are also reported in Tables 11 and 12.

The *p*-value of the *t*-test of most covariates before matching is less than the significant levels of either 0.01, 0.05, or 0.10. This implies that the alternative hypothesis of the mean difference value of each covariate between two groups, being significantly distinct before matching, can be accepted. These differences of both treatment variables are considerably reduced after nearest neighbor matching and kernel matching, because the *p*-value of most covariates become larger than the significant levels of 0.01, 0.05, or 0.10. This implies that the null hypothesis of the equality in mean difference value of each covariate between two groups after matching cannot be rejected. However, only one covariate difference remains. In detail, regarding the propensity score models of both adaptation and CSA, the difference of education covariate alone is statistically significant in terms of nearest neighbor matching.

As expected, the PSM provides appropriate propensity score models and balancing scores. The adequacy and robustness of the estimation of treatment effects and the matching procedure suggest the results have validity.

In summary, the performance of adaptation response and the participation in the CSA pilot program were found to be effective in terms of technical efficiency improvement in rice production. It is worth noting that the analysis using the propensity score matching was accurate and robust in terms of solving the estimation errors by eliminating temporarily invariant sources of selection bias.

**Table 11.** Result of balance test of treatment variable of adaptation response.

| Matching | Covariate | Mean Standardized Bias (%) | | % Reduction Bias | *t*-Test | | *p*-Value | |
|---|---|---|---|---|---|---|---|---|
| | | Unmatched | Matched | | Unmatched | Matched | Unmatched | Matched |
| NNM | Education | 11.2 | 18.8 | −68.3 ** | 0.93 | 2.08 | 0.355 | 0.038 |
| | Experience | −25.7 | 13.0 | 49.2 | −2.21 | 1.42 | 0.027 | 0.157 |
| | Extension | 51.9 | 7.4 | 85.7 | 4.44 | 0.83 | 0.000 | 0.409 |
| | Belief in climate change | 11.9 | −4.5 | 62.3 | 1.02 | −0.51 | 0.307 | 0.607 |
| | Trust in public adaptation | 24.6 | 6.4 | 74.1 | 1.95 | 0.75 | 0.052 | 0.452 |
| | Social norm | 19.2 | −9.6 | 50.1 | 1.61 | −1.05 | 0.108 | 0.295 |
| | Farm area | 24.1 | 8.9 | 63.1 | 1.95 | 0.92 | 0.052 | 0.359 |
| | Region (Long An) | −4.3 | −0.8 | 80.1 | −0.36 | −0.10 | 0.716 | 0.924 |
| | Region (Ben Tre) | −23.2 | 0.8 | 96.4 | −2.01 | 0.10 | 0.045 | 0.922 |
| | Access to water (Near) | −0.5 | 4.8 | −865.6 | −0.04 | 0.54 | 0.966 | 0.589 |
| | Access to water (Medium) | 16.5 | −1.6 | 90.2 | 1.41 | −0.18 | 0.160 | 0.858 |
| KM | Education | 11.2 | 14.9 | −33.3 | 0.93 | 1.53 | 0.355 | 0.126 |
| | Experience | −25.7 | −1.6 | 93.7 | −2.21 | −0.17 | 0.027 | 0.866 |
| | Extension | 51.9 | 5.5 | 89.3 | 4.44 | 0.57 | 0.000 | 0.571 |
| | Belief in climate change | 11.9 | −8.6 | 27.3 | 1.02 | −0.95 | 0.307 | 0.344 |
| | Trust in public adaptation | 19.2 | −11.8 | 38.3 | 1.61 | −1.26 | 0.108 | 0.208 |
| | Social norm | 24.6 | −1.4 | 94.1 | 2.13 | −0.16 | 0.034 | 0.874 |
| | Farm area | 24.1 | 3.0 | 87.7 | 1.95 | 0.32 | 0.052 | 0.747 |
| | Region (Long An) | −4.3 | −1.4 | 67.8 | −0.36 | −0.14 | 0.716 | 0.886 |
| | Region (Ben Tre) | −23.2 | 6.6 | 71.7 | −2.01 | 0.70 | 0.045 | 0.484 |
| | Access to water (Near) | −0.5 | 2.3 | −357.8 | −0.04 | 0.24 | 0.966 | 0.812 |
| | Access to water (Medium) | 16.5 | 0.7 | 95.6 | 1.41 | 0.08 | 0.160 | 0.940 |

Note: ** means significance with confidence interval at 95%. NNM denotes nearest neighbor matching, KM denotes kernel matching.

**Table 12.** Result of balance test of treatment variable of CSA participation.

| Matching | Covariate | Mean Standardized Bias (%) | | % Reduction Bias | t-test | | p-value | |
|---|---|---|---|---|---|---|---|---|
| | | Unmatched | Matched | | Unmatched | Matched | Unmatched | Matched |
| NNM | Education | 30.6 | −59.5 | −94.1 * | 1.35 | −1.84 | 0.178 | 0.072 |
| | Experience | 12.9 | 36.7 | −184.9 | 0.57 | 1.28 | 0.568 | 0.208 |
| | Extension | 110.4 | 11.8 | 89.3 | 3.95 | 0.59 | 0.000 | 0.561 |
| | Belief in climate change | 71.4 | 14.9 | 79.1 | 3.06 | 0.54 | 0.002 | 0.589 |
| | Farm area | 72.2 | −58.0 | 19.7 | 3.55 | −1.06 | 0.000 | 0.295 |
| | Region (Long An) | 16.2 | −9.3 | 42.3 | 0.76 | −0.30 | 0.450 | 0.767 |
| | Region (Ben Tre) | −49.3 | 0.0 | 100.0 | −2.00 | −0.00 | 0.047 | 1.000 |
| | Access to water (Near) | 39.2 | 0.0 | 100.0 | 1.77 | 0.00 | 0.077 | 1.000 |
| | Access to water (Medium) | −32.3 | −9.3 | 71.2 | −1.43 | −0.31 | 0.152 | 0.757 |
| KM | Education | 30.6 | −10.8 | 64.7 | 1.35 | −0.33 | 0.178 | 0.745 |
| | Experience | 12.9 | −1.9 | 85.4 | 0.57 | −0.06 | 0.568 | 0.949 |
| | Extension | 110.4 | 5.9 | 94.6 | 3.95 | 0.28 | 0.000 | 0.782 |
| | Belief in climate change | 71.4 | −0.9 | 98.7 | 3.06 | −0.03 | 0.002 | 0.977 |
| | Farm area | 72.2 | −5.0 | 93.0 | 3.55 | −0.14 | 0.000 | 0.893 |
| | Region (Long An) | 16.2 | 12.7 | 21.6 | 0.76 | 0.38 | 0.450 | 0.706 |
| | Region (Ben Tre) | −49.3 | −0.2 | 99.6 | −2.00 | −0.01 | 0.047 | 0.994 |
| | Access to water (Near) | 39.2 | 0.0 | 100.0 | 1.77 | 0.00 | 0.077 | 1.000 |
| | Access to water (Medium) | −32.3 | −6.7 | 79.1 | −1.43 | −0.21 | 0.152 | 0.835 |

Note: * means significance with confidence interval at 90%. NNM denotes nearest neighbor matching, KM denotes kernel matching.

## 6. Conclusions

The study attempted to examine the factors affecting the performance of CSA and climate change adaptation response and associated effects on technical efficiency of rice farming in the Mekong Delta of Vietnam.

The findings reported that, to cope with climate change effects related to salinity intrusion and drought, most farmers (71%) performed their adaptation response by individual or collective practices. Others (29%) performed no adaptation response. Additionally, twenty-two rice farmers were typically chosen as participants in the CSA pilot program organized by local government and agricultural materials companies.

In determining the decisional factors driving decisions on adaptation response and CSA participation and assessing the effects of adaptation performance and CSA participation on technical efficiency of rice production, the PSM approach was recruited. The empirical results, obtained from the propensity score models, indicated that some key factors affecting the performance of adaptation response and CSA participation were comprised of agricultural extension services, belief in climate change, the area of farming land, and geographical location, such as provinces and access to water sources.

This study's findings support the main role of climate change adaptation response and CSA in improving the technical efficiency of rice farming. Farmers who have adapted their rice farming to climate change effects related to salinity intrusion and drought had a 13%–14% higher technical efficiency than those who have not adapted. Specifically, participation in the CSA pilot program also yielded a 5%–8% higher technical efficiency compared to no participation.

In addition, the PSM was found to accurately estimate the treatment effects of adaptation response and CSA after eliminating the problem of selection bias.

These results have important policy implications in terms of incentives to strongly support adaptation response, by providing precise and timely information associated with climate change adaptation practices and CSA suitable for geographical location through agricultural extension services at the local level. In addition, popularizing local adaptation practices and the CSA program, due to it achievements on economic performance (i.e., technical efficiency), is also necessary.

The authors hope this empirical study contributes to an approach for estimating the impacts of adaptive behavior in response to climate change effects associated with salinity intrusion and drought.

**Author Contributions:** T.T.H. and K.S. organized the research design, implemented the field survey, and analyzed the data; T.T.H. mainly wrote the paper.

**Funding:** This research was funded by Social System Research Institute Mission, Ritsumeikan University in the year of 2018.

**Acknowledgments:** We appreciate the support of the Department of Agriculture and Rural Development at six districts (Tan Thanh and Can Duoc (Long An province), Thanh Phu and Ba Tri (Ben Tre province), and Tieu Can and Tra Cu (Tra Vinh province)) for their help in organizing farmer interviews and provision of agricultural information and secondary data. We also thank my colleagues Pham Thi Anh Ngoc, Tran Hoai Nam, and Le Vu, and six undergraduate students at Nong Lam University, Ho Chi Minh City, for their help in field survey preparation and data collection.

**Conflicts of Interest:** The authors declare no conflict of interest.

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
