# Peer review of "The Effects of Climate Smart Agriculture and Climate Change Adaptation on the Technical Efficiency of Rice Farming—An Empirical Study in the Mekong Delta of Vietnam"

_agriculture, doi:10.3390/agriculture9050099_

Reviewer 1 Report

General

×        The paper’s findings support idea that CSA practices improve technical efficiency in rice farming, which is a useful finding.

×        The paper also has useful policy implications, namely underlining how CSA measures can improve both resilience to CC and technical efficiency. This includes suggesting a need to support adaptation responses via different mechanisms, e.g., information production via ag extension, and popularising CSA programme and its effects on economic performance.

×        The data gathered and relationships examined seem worthwhile, and it would be good if these were reported via an academic paper. This could be genuinely useful, since providing evidence on the economic benefits of CSA practices is needed to support wider adoption of these practices by farmers. This being said, reporting of such findings needs to be done in a serious, convincing way, and unfortunately this paper does not achieve this at present.

×        One major problem is that the language used is confused in various places. It would be good if the authors could work with someone with strong English to improve paper’s narrative and argument. As things stand, various sentences are confused, and confusing to the reader. The content of the argument is also problematic in places.

×        The authors need to go back and think about their task and make a rigorous argument if they wish to resubmit this paper for publication.

Comments on specific sections of the text

×        39-44: This paragraph starts with noting that ag is a significant contributor to CC, but then switches to ways that CC undermines agriculture. This is OK, but the paragraph needs to begin with a neutral framing of how agriculture is both a cause of CC and a sector that is adversely impacted by CC impacts.

×        45-48: The characterisation of CSA is incomplete. The authors should read basic definitions of CSA, for instance from the United Nations FAO.

×        57-60: Paper covers an important theme, namely the significance of CSA practices to rice farming in Vietnam, including via maintaining competitiveness and exports via enhanced technical efficiency.

×        62-64: The use of the term ‘mitigation’ is incorrect. They are using it in the sense of disaster risk management or colloquial English. But this is completely different than the meaning of ‘mitigation’ in the context of climate change.

×        67-68: The focus of the paper is good, as this is an important theme. This sentence summarises this focus, but even this critical sentence is garbled. “This study focuses on determining the effects of climate change adaptation response and CSA on technical efficiency of rice farming at farm level in the Vietnamese Mekong.” In fact, the study focuses on the effect of CSA practices on the technical efficiency of rice farming, and also on its resilience to climate change. The authors really need to say this clearly.

×        75-79: The authors cite an example of evidence from Pakistan without including a reference. This is just plain sloppy and unserious.

×        81-86: Again, the key word in this sentence is ‘mitigate’, and yet the authors have misused this term. This is not the usage associated with work on climate change. They clearly do not understand that ‘adaptation’ and ‘mitigation’ refer to wholly distinct concepts.

×        87-91: Here the authors have used the term ‘mitigation’ correctly, but this does not fit with their earlier uses of this term. This will only confuse the reader, in addition to being a clear mistake.

×        125-27: This is an odd conclusion. “Therefore the need of evaluating the effects of adaptation response and CSA participation on technical efficiency... becomes urgent”. It seems quite possible that the authors are confusing the terms ‘adoption’ and ‘adaptation’.  

×        128-129: From here, the authors talk at length about using a certain approach to PSM “to overcome the problem of selection bias when using observational data”, but it’s not clear why selection bias when using observational data is a problem here. To explain this, they should briefly explain the problem, then explain why they chose this approach instead of others. For instance, they could perhaps have simply used statistical sampling, or some variant thereof such as probability proportional to size.

×        251: Unclear what “the more pressures” means.

×        268: Is it legitimate for participation in the CSA pilot programme as a dependent variable? It doesn’t seem to be, since in 310-311, the authors tell us that farmers “are typically chosen by local government and institutions for participation”. So this tells us nothing about their actual behaviour or decisions. If non-participants have lower technical efficiency, this simply suggest that government agents seek to select farmers they deem more successful, thus showing a selection bias. This is not science, but rather favouritism and trying to make the local area look good by showcasing ‘winners’. While it is understandable for government agents to wish to showcase ‘winners’, it is not acceptable for scientists to treat such differences between participants and non-participants as evidence of for identifying determinants of technical efficiency.

Author Response

Response to Reviewer 1 Comments

We appreciate for your helpful comments and suggestions very much. We have indicated all required changes in red color, all grammar check highlighted in yellow color, and all the other changes highlighted in grey color in the revised manuscript. We have revised the manuscript according to your comments and suggestions point by point as follows.

In terms of general comments

Response:

We have improved the use of language by grammar check (all highlighted in yellow color).

-        We added ‘the’ before ‘technical efficiency’ in the 15th line of page 1.

-        We deleted ‘of’ between ‘observational data’ and ‘were’ in the 15th line of page 1.

-        We added ‘the’ between ‘in’ and ‘agriculture’ in the 32nd line of page 1

-        We added ‘production’ after ‘grain food’ in the 33rd line of page 1.

-        We change ‘is’ to ‘was’ in the 33rd line of page 1.

-        We deleted ‘which’, added ‘,’ after ‘4.88 million tons’ in the 33rd line of page 1, and changed ‘accounts’ to ‘accounting’ in the 34th line of page 1.

-        We deleted ‘the’ before ‘Vietnam’s Mekong Delta’ in the 34th line of page 1.

-        We changed ‘contribute’ to ‘contributes’ in the 35th line of page 1.

-        We replaced ‘surface’ by ‘storage’ in the 42nd line of page 1.

-        We deleted ‘of’ between ‘most’ and ‘regions’ in the 44th line of page 2.

-        We added ‘the’ before ‘national economy’ in the 57th line pf page 2.

-        We added ‘the’ before ‘case of implementing CSA’ in the 63rd-64th line of page 2.

-        We changed ‘CSA’ to ‘the CSA program’ in the 67th line of page 2.

-        We added ‘the’ before ‘agricultural sector’ in the 72nd line of page 2.

-        We replaced ‘crop season’ by ‘crop plantings’ in the 83rd line of page 2.

-        We added ‘the’ before ‘agricultural production system’ in the 89th line of page 2.

-        We added ‘the’ before ‘agricultural sector’ in the 91st line of page 2.

-        We deleted ‘the’ before ‘sustainable agricultural intensification practices’ in the 92nd line of page 2.

-        We changed ‘fertilizer company, chemical company’ to ‘fertilizer companies, chemical companies’ in the 98th line of page 3.

-        We added ‘the’ before ‘Mekong Delta’ in the 101st line of page 3.

-        We replaced ‘the’ by ‘a’ in the 101st line of page 3.

-        We changed ‘government’ to ‘governments’, ‘provided’ to ‘supported’ in the 105th line of page 3.

-        We changed ‘technique’ to ‘techniques’, ‘variety’ to ‘varieties’, and ‘chemical’ to ‘chemicals’ in the 105th-106th line of page 3.

-        We added ‘the’ before ‘propensity score matching’ in the 111th line of page 3.

-        We added ‘the’ before ‘agricultural sector’ in the 115th line of page 3.

-        We added ‘the’ before ‘number of adaptation measures’ in the 117th line of page 3.

-        We added ‘the’ before ‘Eastern and Central provinces of Kenya’ in the 124h line of page 3.

-        We changed ‘farm’ to ‘farming’ in the 124th line of page 3.

-        We added ‘the’ before ‘matching procedure’ in the 166th line of page 4.

-        We added ‘the’ before ‘two main methods’ in the 175th line of page 4.

-        We replaced ‘with’ by ‘at’ in the 210th line of page 4.

-        We replaced ‘high’ by ‘elevation’ in the 211st line of page 4.

-        We added ‘elevation’ after ‘0.3 to 0.7m’ in the 211st line of page 4.

-        We added ‘this’ between ‘due to’ and ‘low topography’ in the 211st line of page 4.

-        We added ‘the’ before ‘East sea’ in the 212nd line of page 4.

-        We added ‘the’ before ‘dry season’ in the 213rd line of page 4.

-        We changed ‘province’ to ‘provinces’ in the 216th and 217th line of page 5.

-        We added ‘upon’ between ‘intruded’ and ‘by’ in the 220th line of page 5.

-        We added ‘only’ between ‘are’ and ‘slightly’ in the 222nd line of page 5.

-        We replaced ‘withdrawn’ by ‘picked up’ in the 224th line of page 5.

-        We added ‘the’ before ‘Mekong Delta’ in the 226th line of page 5.

-        We replaced ‘including’ by ‘, more specifically’ in the 226th line of page 5.

-        We replaced ‘guides’ by ‘guidance’ in the 229th line of page 5.

-        We replaced ‘seen’ by ‘saw’ in the 251th line of page 6.

-        We change ‘positively’ to ‘a positive one’ in the 255th line of page 6.

-        We deleted ‘,’ and added ‘and’ between ‘education level’ and ‘farming experience’ in the 258th line of page 6.

-        We changed ‘cognitive factor’ to ‘cognitive factors’ and ‘is’ to ‘are’ in the 261st line of page 6.

-        We added ‘the’ before ‘provincial level’ in the 264th line of page 6, the 274th line of page 7, and the 289th line of page 8.

-        We added ‘more’ before ‘farmers’ in the 279th line of page 7.

-        We added ‘the’ before ‘CSA pilot program’ in the 290th line of page 8.

-        We added ‘a’ before ‘higher level of education’ in the 291st line of page 8.

-        We added ‘by a tendency’ between ‘explained’ and ‘that’ in the 292nd line of page 8.

-        We changed ‘near’ to ‘nearer’ in the 297th line of page 8.

-        We added ‘the’ before ‘rice farms’ in the 298th line of page 8.

-        We added ‘more often’ between ‘are’ and ‘located’ in the 298th line of page 8.

-        We changed ‘change rice variety’ to ‘and changing rice variety’ in the 316th and 319th line of page 9.

-        We added ‘the’ before ‘CSA program’ in the 317th and 318th line of page 9.

-        We changed ‘improving crop’ to ‘improving crops’ in the 322nd line of page 9.

-        We replaced ‘crop season’ by ‘crop plantings’ in the 323rd line of page 9 and the 334th line of page 10.

-        We added ‘the’ before ‘area of farming land’ in the 325th line of page 9.

-        We added ‘More specifically,’ before ‘all four input variables’ in the 339th line of page 10.

-        We replaced ‘by 15.29%’ by ‘a 15.29%’ before ‘lower technical efficiency’ in the 348th line of page 10.

-        We replaced ‘by 12.07%’ by ‘a 12.07%’ before ‘lower technical efficiency’ in the 356th line of page 10.

-        We added ‘a’ before ‘micro-level’ in the 380th line of page 12.

-        We added ‘of’ between ‘are’ and ‘statistically positive significance’ in the 389th line of page 12.

-        We replaced ‘by 13.57 per cent’ by ‘a 13.57%’ before ‘higher technical efficiency’ in the 393rd line of page 12.

-        We replaced ‘by 12.66 per cent’ by ‘a 12.66%’ before ‘higher technical efficiency’ in the 395th line of page 12.

-        We replaced ‘by 4.51 per cent’ by ‘a 4.51%’ before ‘higher technical efficiency’ in the 408th-409th line of page 13.

-        We replaced ‘by 8.2 per cent’ by ‘a 8.2%; before ‘higher technical efficiency’ in the 410th line of page 13.

-        We added ‘a’ before ‘possible reason’ in the 411th line of page 13.

-        We added ‘the’ before ‘CSA program’ in the 411th line of page 13.

-        We changed ‘adaptation response’ to ‘adaptation responses’ in the 414th line of page 13.

-        We replaced ‘crop season’ by ‘crop plantings’ in the 416th line of page 13.

-        We changed ‘diversify crop’ to ‘diversify crops’ in the 416th line of page 13.

-        We added ‘the’ before ‘technical efficiency’ in the 487th line of page 18.

-        We replaced ‘by 13-14 per cent’ by ‘a 13-14%’ before ‘higher technical efficiency’ in the 488th-489th line of page 18.

-        We added ‘the’ before ‘CSA pilot program’ in the 489th line of page 18.

-        We changed ‘yield’ to ‘yields’ in the 490th line of page 18.

-        We replaced ‘by 5-8 per cent’ by ‘a 5-8%’ before ‘higher technical efficiency’ in the 490th line of page 18.

-        We changed ‘authors hope this empirical study be able to provide contribution into’ to ‘the author hopes this empirical study contributes to an’

-        We added into the part of Funding the following sentence ‘This research was funded by Social System Research Institute Mission, Ritsumeikan University in the year of 2018.”

-        We changed “Giuoc” to “Duoc” in the 508th line of page 18.

-        We changed ‘Agricultue’ to ‘Agriculture’ in the 516th line of page 19.

-        We changed ‘eficiency’ to ‘efficiency’ in the 537th line of page 19.

Additionally, we have changed the structure of the content (all highlighted in grey color) as follows:

-        We changed the section 4 of Study site and Data Description into the section 3 of Study site and Data collection which includes Study site (3.1) and Data collection (3.2).

-        We moved the part of Data description (4.3) into the section 4 of Methodological framework which includes Methodological framework (4.1) and Data description (4.2).

-        We edited references and changed all the order of following references in the whole main text. The section of Reference has been changed as follows:

Ø  We added reference 4 after the explanation of how agriculture is a sector that is adversely impacted by climate change impacts from the 40th line of page 1 as follows:

“OECD. Meeting of Agriculture Ministers – Background note.  Available online: https://www.oecd.org/agriculture/ministerial/background/notes/4_background_note.pdf (accessed on 17 April 2019).”

Ø  Reference ‘4’ to ‘5’, ‘5’ to ‘6’, ‘6’ to ‘7’, ‘7’ to ‘8’, ‘8’ to ‘9’, ‘9’ to ‘10’, ‘10’ to ‘11’, ‘11’ to ‘12’, ‘12’ to ‘13’

Ø  Reference ‘13’ to ‘15’

Ø  We added reference 16 after the sentence ‘This region has been reported as significantly vulnerable to climate change’ in the 208th line of page 5 as follows:

“Yusuf, A.A., Francisco, H. Climate change vulnerability mapping in Southeast Asia, 2010. Available online: https://www.idrc.ca/sites/default/files/sp/Documents%20EN/climate-change-vulnerability-mapping-sa.pdf (Accessed on 10 September 2018).”

Ø  Reference ‘36’ to ‘17’

Ø  Reference ‘15’ to ‘18’,

Ø  Reference ‘24’ to ‘19’

Ø  Reference ‘16’ to ‘20’, ‘17’ to ‘21’, ‘18’ to ‘22’, ‘19’ to ‘23’, ‘20’ to ‘24’, ‘21’ to ‘25’, ‘22’ to ‘26’, ‘23’ to ‘27’, ‘25’ to ‘28’, ‘26’ to ‘29’, ‘27’ to ‘30’, ‘28’ to ‘31’, ‘29’ to ‘32’, ‘30’ to ’33’, ‘31’ to ‘34’, ‘32’ to ‘35’, ‘33’ to ‘36’, ‘34’ to ‘37’, ‘35’ to ‘38’, ‘37’ to ‘39’, ‘38’ to ‘40’, ‘39’ to ‘41’, ‘40’ to ‘42’, ‘41’ to ‘43’, and ‘42’ to ‘44’.

Ø  We added the volume and pages of the journal into reference 44 as follows:

“2019, 67, 130-144”

-        We have replaced the Figure 2 related to Adaptation practices implemented by local rice farmers (after added Figure 1 related to map of study areas).

In terms of specific comments

1.     39-44: This paragraph starts with noting that ag is a significant contributor to CC, but then switches to ways that CC undermines agriculture. This is OK, but the paragraph needs to begin with a neutral framing of how agriculture is both a cause of CC and a sector that is adversely impacted by CC impacts.

Response 1:

We agreed to the comment and have added the explanation of how agriculture is a sector that is adversely impacted by climate change impacts as follows:

“On the other hand, agriculture is defined as a sector that heavily depend on natural resources and climate, and thus may suffer the adverse impacts of climate change. Specifically, climate change related to increases in temperature, precipitation variation, and the frequency and intensity of extreme weather events are putting pressures on the agricultural systems [4].”

2.     45-48: The characterization of CSA is incomplete. The authors should read basic definitions of CSA, for instance from the United Nations FAO.

Response 2:

We have added a brief definition of CSA in this part because the definition of CSA was fully described in the section 2. We have changed the brief definition of CSA as follows:

“To enhance the resilience of the agriculture, reduce greenhouse gas (GHG) emissions, and manage input resources for a sustainable agriculture under the challenges of climate change, climate smart agriculture, which interlinks with adaptation and mitigation is considered as an important approach, particularly in developing country.”

à “Furthermore, to adapt or enhance the resilience of the agriculture, reduce greenhouse gas (GHG) emissions, and manage input resources for a sustainable agriculture under the challenges of climate change, climate smart agriculture, which integrates the three dimensions of sustainable development (social, environmental, and economic) is considered as an important approach to achieve food security and agricultural development goals, particularly in developing countries.”

In addition, to make clear the role of adaptation, we have added the sentence “To cope with these risks, farmers decided to perform their adaptation response by adjusting or adapting their farming practices” before the sentence “More specifically, adaptation is pressing to reduce the vulnerability to the adverse impacts of climate change,…” and this paragraph was moved after the paragraph of “However, agriculture is also a major part of the problem related to global warming and climate change… in the Mekong Delta” in the 39th-44th line.

3.     57-60: Paper covers an important theme, namely the significance of CSA practices to rice farming in Vietnam, including via maintaining competitiveness and exports via enhanced technical efficiency.

Response 3:

We would like to thank you very much for your positive comment.

4.     62-64: The use of the term ‘mitigation’ is incorrect. They are using it in the sense of disaster risk management or colloquial English. But this is completely different than the meaning of ‘mitigation’ in the context of climate change.

Response 4:

We have changed the term ‘mitigated’ to ‘moderated’ in the 63th line of page 2.

5.      67-68: The focus of the paper is good, as this is an important theme. This sentence summarises this focus, but even this critical sentence is garbled. “This study focuses on determining the effects of climate change adaptation response and CSA on technical efficiency of rice farming at farm level in the Vietnamese Mekong.” In fact, the study focuses on the effect of CSA practices on the technical efficiency of rice farming, and also on its resilience to climate change. The authors really need to say this clearly.

Response 5:

Although building resilience to climate change is important to cope with climate change, the study only focuses on estimating the effects of climate change adaptation practices and CSA program on the technical efficiency of rice farming with a limitation in estimating the effects of CSA program and climate change adaptation practices on its resilience to climate change. Therefore, we would like to keep this sentence “This study focuses on determining the effects of climate change adaptation response and the CSA program on technical efficiency of rice farming at farm level in the Vietnamese Mekong.”

6.     75-79: The authors cite an example of evidence from Pakistan without including a reference. This is just plain sloppy and unserious.

Response 6:

We have deleted the citation of references [7-9,13] after the sentence “The implementation of adaptation practices can not only help rural farmers minimize potential damages in crop yield but also sustain their income and food security status” and cited each example with a reference as follows:

“In details, the adoption of improved maize in Eastern Zambia contributes to significant gains in income (by 78,900 ZMK per ha), consumption expenditure (by 324,690 ZMK per capita), and food security (by 2%) as well as significant reduction in poverty (by 21%) [8].”

“Adaptation practices in four main provinces of Pakistan had substantial effects on improving net income by 1,658 – 2,610 Pakistani rupee per month and increasing wheat yield by 42 – 65 kg per hectare [9].”

“Moreover, adaptation practices taken by Pakistani farmers had positively significant effects on food security level (8 – 13%) and negatively significant effects on poverty level (3 – 6%) [10].”

In addition, we have added one more example of evidence from rural Zambia with a reference citation “ Furthermore, the adoption of sustainable agricultural practices in maize farming involving the promotion of conservation agriculture package (e.g. crop residue retention, maize-legume rotation) as well as improved crop varieties played a crucial role in raising yield and income of smallholder farmers in rural Zambia [15].” after the sentence “To increase soil productivity,…by 11 percent” in the 93th line of page 2.

7.     81-86: Again, the key word in this sentence is ‘mitigate’, and yet the authors have misused this term. This is not the usage associated with work on climate change. They clearly do not understand that ‘adaptation’ and ‘mitigation’ refer to wholly distinct concepts.

Response 7:

We have changed the term ‘mitigate’ to ‘cope’ in the 85th line of page 2.

8.     87-91: Here the authors have used the term ‘mitigation’ correctly, but this does not fit with their earlier uses of this term. This will only confuse the reader, in addition to being a clear mistake.

Response 8:

According to your suggestion, we have clarified and modified the term ‘mitigate’ and ‘mitigation’ in the earlier use.

9.     25-27: This is an odd conclusion. “Therefore, the need of evaluating the effects of adaptation response and CSA participation on technical efficiency... becomes urgent”. It seems quite possible that the authors are confusing the terms ‘adoption’ and ‘adaptation’.

Response 9:

We used the term of adaptation response throughout the study which explain the ways farmers adapted their farming in response to climate change impacts. Meanwhile, the term of adoption was only used in examples from references where authors mentioned the adoption of specific measure to cope with climate change. Therefore, we would like to keep the term of adaptation response.

Otherwise, we have changed this sentence more clearly as follows:

“Therefore, the need of evaluating the effects of adaptation response and CSA participation on technical efficiency of rice farming in the Mekong Delta, Vietnam becomes urgent.”

à“Therefore, this study attempts to evaluate the effects of climate change adaptation response and the CSA program on the technical efficiency of rice farming in the Mekong Delta, Vietnam using a PSM approach.”

10.  128-129: From here, the authors talk at length about using a certain approach to PSM “to overcome the problem of selection bias when using observational data”, but it’s not clear why selection bias when using observational data is a problem here. To explain this, they should briefly explain the problem, then explain why they chose this approach instead of others. For instance, they could perhaps have simply used statistical sampling, or some variant thereof such as probability proportional to size.

Response 10:

We agreed with the comment and have added the explanation of the problem and the choice of PSM approach in the first paragraph of the section of methodological framework as follows:

“Both interviews and CSA pilot program in this study were assigned by administrator selection with lack of randomization. This could face a problem of selection bias that may lead to not only endogeneity bias or confoundedness but also bias estimation of causal effects. Therefore, it is important to apply the propensity score matching approach proposed by Rosenbaum and Rubin [19] for adjusting selection bias as well as balancing data in terms of control covariance before evaluating the treatment effects.

In addition, we have deleted the sentence “To overcome the problem of selection bias when using observational data, the study uses the PSM approach proposed by Rosenbaum and Rubin [24].”

Furthermore, after the sentence in the 148th line of page 4 “After the propensity scores are predicted by the propensity score model,…in the second step.”, we have added a sentence “More specifically, a pair of treated and control subjects sharing same propensity scores are essential considered as comparable for matching, even though they still could differ on specific observed covariates.”

11.  251: Unclear what “the more pressures” means.

Response 11:

We have changed ‘the more pressure’ to ‘the higher the pressure’ in the 251th line of page 6.

12.  268: Is it legitimate for participation in the CSA pilot programme as a dependent variable? It doesn’t seem to be, since in 310-311, the authors tell us that farmers “are typically chosen by local government and institutions for participation”. So this tells us nothing about their actual behaviour or decisions. If non-participants have lower technical efficiency, this simply suggest that government agents seek to select farmers they deem more successful, thus showing a selection bias. This is not science, but rather favouritism and trying to make the local area look good by showcasing ‘winners’. While it is understandable for government agents to wish to showcase ‘winners’, it is not acceptable for scientists to treat such differences between participants and non-participants as evidence of for identifying determinants of technical efficiency.

Response 12:

In this situation, we think the participation in the CSA pilot program as a dependent variable. We have explained in the detail the assignment of the program participation.

-        We have replaced ‘by local government and institutions for participation’ by ‘as participants’ in the line 310th-311th of the page 9.

-        In addition, we have clarified the assignment of CSA participation in the end of section 2 and have added the sentences as follows:

In terms of assigning participants, several local rice farmers (introduced by local officers based on the official list of rice farm household in specific communes) were invited to an agricultural extension service training organized by local government and institutions. Then, they were introduced in detailed about CSA pilot program as well as were asked to voluntarily join. This means that administrators made decisions to recruit local farmers to an introductory training of CSA pilot program while farmers made decisions to participate in the program or not based on their intentions.”

Reviewer 2 Report

I think this paper is fine to be accepted after it has been through a thorough English grammar check.

I found the characteristics considered to be relevant and interesting, and although the scope of adaptation to climate change was mainly limited to salinity and drought the findings and approach are still relevant to wider climate change attributes.

Author Response

Response to Reviewer 2 Comments

Comments and Suggestions for Authors

I think this paper is fine to be accepted after it has been through a thorough English grammar check.

I found the characteristics considered to be relevant and interesting, and although the scope of adaptation to climate change was mainly limited to salinity and drought the findings and approach are still relevant to wider climate change attributes.

Response:

We would like to thank you very much for your valuable comments. We have indicated all required changes in red color, all grammar check highlighted in yellow color, and all the other changes highlighted in grey color in the revised manuscript.

Firstly, we have improved the use of language by grammar check (all highlighted in yellow color).

-        We added ‘the’ before ‘technical efficiency’ in the 15th line of page 1.

-        We deleted ‘of’ between ‘observational data’ and ‘were’ in the 15th line of page 1.

-        We added ‘the’ between ‘in’ and ‘agriculture’ in the 32nd line of page 1

-        We added ‘production’ after ‘grain food’ in the 33rd line of page 1.

-        We change ‘is’ to ‘was’ in the 33rd line of page 1.

-        We deleted ‘which’, added ‘,’ after ‘4.88 million tons’ in the 33rd line of page 1, and changed ‘accounts’ to ‘accounting’ in the 34th line of page 1.

-        We deleted ‘the’ before ‘Vietnam’s Mekong Delta’ in the 34th line of page 1.

-        We changed ‘contribute’ to ‘contributes’ in the 35th line of page 1.

-        We replaced ‘surface’ by ‘storage’ in the 42nd line of page 1.

-        We deleted ‘of’ between ‘most’ and ‘regions’ in the 44th line of page 2.

-        We added ‘the’ before ‘national economy’ in the 57th line pf page 2.

-        We added ‘the’ before ‘case of implementing CSA’ in the 63rd-64th line of page 2.

-        We changed ‘CSA’ to ‘the CSA program’ in the 67th line of page 2.

-        We added ‘the’ before ‘agricultural sector’ in the 72nd line of page 2.

-        We replaced ‘crop season’ by ‘crop plantings’ in the 83rd line of page 2.

-        We added ‘the’ before ‘agricultural production system’ in the 89th line of page 2.

-        We added ‘the’ before ‘agricultural sector’ in the 91st line of page 2.

-        We deleted ‘the’ before ‘sustainable agricultural intensification practices’ in the 92nd line of page 2.

-        We changed ‘fertilizer company, chemical company’ to ‘fertilizer companies, chemical companies’ in the 98th line of page 3.

-        We added ‘the’ before ‘Mekong Delta’ in the 101st line of page 3.

-        We replaced ‘the’ by ‘a’ in the 101st line of page 3.

-        We changed ‘government’ to ‘governments’, ‘provided’ to ‘supported’ in the 105th line of page 3.

-        We changed ‘technique’ to ‘techniques’, ‘variety’ to ‘varieties’, and ‘chemical’ to ‘chemicals’ in the 105th-106th line of page 3.

-        We added ‘the’ before ‘propensity score matching’ in the 111th line of page 3.

-        We added ‘the’ before ‘agricultural sector’ in the 115th line of page 3.

-        We added ‘the’ before ‘number of adaptation measures’ in the 117th line of page 3.

-        We added ‘the’ before ‘Eastern and Central provinces of Kenya’ in the 124h line of page 3.

-        We changed ‘farm’ to ‘farming’ in the 124th line of page 3.

-        We added ‘the’ before ‘matching procedure’ in the 166th line of page 4.

-        We added ‘the’ before ‘two main methods’ in the 175th line of page 4.

-        We replaced ‘with’ by ‘at’ in the 210th line of page 4.

-        We replaced ‘high’ by ‘elevation’ in the 211st line of page 4.

-        We added ‘elevation’ after ‘0.3 to 0.7m’ in the 211st line of page 4.

-        We added ‘this’ between ‘due to’ and ‘low topography’ in the 211st line of page 4.

-        We added ‘the’ before ‘East sea’ in the 212nd line of page 4.

-        We added ‘the’ before ‘dry season’ in the 213rd line of page 4.

-        We changed ‘province’ to ‘provinces’ in the 216th and 217th line of page 5.

-        We added ‘upon’ between ‘intruded’ and ‘by’ in the 220th line of page 5.

-        We added ‘only’ between ‘are’ and ‘slightly’ in the 222nd line of page 5.

-        We replaced ‘withdrawn’ by ‘picked up’ in the 224th line of page 5.

-        We added ‘the’ before ‘Mekong Delta’ in the 226th line of page 5.

-        We replaced ‘including’ by ‘, more specifically’ in the 226th line of page 5.

-        We replaced ‘guides’ by ‘guidance’ in the 229th line of page 5.

-        We replaced ‘seen’ by ‘saw’ in the 251th line of page 6.

-        We change ‘positively’ to ‘a positive one’ in the 255th line of page 6.

-        We deleted ‘,’ and added ‘and’ between ‘education level’ and ‘farming experience’ in the 258th line of page 6.

-        We changed ‘cognitive factor’ to ‘cognitive factors’ and ‘is’ to ‘are’ in the 261st line of page 6.

-        We added ‘the’ before ‘provincial level’ in the 264th line of page 6, the 274th line of page 7, and the 289th line of page 8.

-        We added ‘more’ before ‘farmers’ in the 279th line of page 7.

-        We added ‘the’ before ‘CSA pilot program’ in the 290th line of page 8.

-        We added ‘a’ before ‘higher level of education’ in the 291st line of page 8.

-        We added ‘by a tendency’ between ‘explained’ and ‘that’ in the 292nd line of page 8.

-        We changed ‘near’ to ‘nearer’ in the 297th line of page 8.

-        We added ‘the’ before ‘rice farms’ in the 298th line of page 8.

-        We added ‘more often’ between ‘are’ and ‘located’ in the 298th line of page 8.

-        We changed ‘change rice variety’ to ‘and changing rice variety’ in the 316th and 319th line of page 9.

-        We added ‘the’ before ‘CSA program’ in the 317th and 318th line of page 9.

-        We changed ‘improving crop’ to ‘improving crops’ in the 322nd line of page 9.

-        We replaced ‘crop season’ by ‘crop plantings’ in the 323rd line of page 9 and the 334th line of page 10.

-        We added ‘the’ before ‘area of farming land’ in the 325th line of page 9.

-        We added ‘More specifically,’ before ‘all four input variables’ in the 339th line of page 10.

-        We replaced ‘by 15.29%’ by ‘a 15.29%’ before ‘lower technical efficiency’ in the 348th line of page 10.

-        We replaced ‘by 12.07%’ by ‘a 12.07%’ before ‘lower technical efficiency’ in the 356th line of page 10.

-        We added ‘a’ before ‘micro-level’ in the 380th line of page 12.

-        We added ‘of’ between ‘are’ and ‘statistically positive significance’ in the 389th line of page 12.

-        We replaced ‘by 13.57 per cent’ by ‘a 13.57%’ before ‘higher technical efficiency’ in the 393rd line of page 12.

-        We replaced ‘by 12.66 per cent’ by ‘a 12.66%’ before ‘higher technical efficiency’ in the 395th line of page 12.

-        We replaced ‘by 4.51 per cent’ by ‘a 4.51%’ before ‘higher technical efficiency’ in the 408th-409th line of page 13.

-        We replaced ‘by 8.2 per cent’ by ‘a 8.2%; before ‘higher technical efficiency’ in the 410th line of page 13.

-        We added ‘a’ before ‘possible reason’ in the 411th line of page 13.

-        We added ‘the’ before ‘CSA program’ in the 411th line of page 13.

-        We changed ‘adaptation response’ to ‘adaptation responses’ in the 414th line of page 13.

-        We replaced ‘crop season’ by ‘crop plantings’ in the 416th line of page 13.

-        We changed ‘diversify crop’ to ‘diversify crops’ in the 416th line of page 13.

-        We added ‘the’ before ‘technical efficiency’ in the 487th line of page 18.

-        We replaced ‘by 13-14 per cent’ by ‘a 13-14%’ before ‘higher technical efficiency’ in the 488th-489th line of page 18.

-        We added ‘the’ before ‘CSA pilot program’ in the 489th line of page 18.

-        We changed ‘yield’ to ‘yields’ in the 490th line of page 18.

-        We replaced ‘by 5-8 per cent’ by ‘a 5-8%’ before ‘higher technical efficiency’ in the 490th line of page 18.

-        We changed ‘authors hope this empirical study be able to provide contribution into’ to ‘the author hopes this empirical study contributes to an’

-        We added into the part of funding the following sentence ‘This research was funded by Social System Research Institute Mission, Ritsumeikan University in the year of 2018.”

-        We changed “Giuoc” to “Duoc” in the 508th line of page 18.

-        We changed ‘Agricultue’ to ‘Agriculture’ in the 516th line of page 19.

-        We changed ‘eficiency’ to ‘efficiency’ in the 537th line of page 19.

Additionally, we have changed the structure of the content (all highlighted in grey color) as follows:

-        We changed the section 4 of Study site and Data Description into the section 3 of Study site and Data collection which includes Study site (3.1) and Data collection (3.2).

-        We moved the part of Data description (4.3) into the section 4 of Methodological framework which includes Methodological framework (4.1) and Data description (4.2).

-        We edited references and changed all the order of following references in the whole main text. The section of Reference has been changed as follows:

Ø  We added reference 4 after the explanation of how agriculture is a sector that is adversely impacted by climate change impacts from the 40th line of page 1 as follows:

“OECD. Meeting of Agriculture Ministers – Background note.  Available online: https://www.oecd.org/agriculture/ministerial/background/notes/4_background_note.pdf (accessed on 17 April 2019).”

Ø  Reference ‘4’ to ‘5’, ‘5’ to ‘6’, ‘6’ to ‘7’, ‘7’ to ‘8’, ‘8’ to ‘9’, ‘9’ to ‘10’, ‘10’ to ‘11’, ‘11’ to ‘12’, ‘12’ to ‘13’

Ø  Reference ‘13’ to ‘15’

Ø  We added reference 16 after the sentence ‘This region has been reported as significantly vulnerable to climate change’ in the 208th line of page 5 as follows:

“Yusuf, A.A., Francisco, H. Climate change vulnerability mapping in Southeast Asia, 2010. Available online: https://www.idrc.ca/sites/default/files/sp/Documents%20EN/climate-change-vulnerability-mapping-sa.pdf (Accessed on 10 September 2018).”

Ø  Reference ‘36’ to ‘17’

Ø  Reference ‘15’ to ‘18’,

Ø  Reference ‘24’ to ‘19’

Ø  Reference ‘16’ to ‘20’, ‘17’ to ‘21’, ‘18’ to ‘22’, ‘19’ to ‘23’, ‘20’ to ‘24’, ‘21’ to ‘25’, ‘22’ to ‘26’, ‘23’ to ‘27’, ‘25’ to ‘28’, ‘26’ to ‘29’, ‘27’ to ‘30’, ‘28’ to ‘31’, ‘29’ to ‘32’, ‘30’ to ’33’, ‘31’ to ‘34’, ‘32’ to ‘35’, ‘33’ to ‘36’, ‘34’ to ‘37’, ‘35’ to ‘38’, ‘37’ to ‘39’, ‘38’ to ‘40’, ‘39’ to ‘41’, ‘40’ to ‘42’, ‘41’ to ‘43’, and ‘42’ to ‘44’.

Ø  We added the volume and pages of the journal into reference 44 as follows:

“2019, 67, 130-144”

-        We have replaced the Figure 2 related to Adaptation practices implemented by local rice farmers (after added Figure 1 related to map of study areas)

Reviewer 3 Report

The paper aims to assess the effects of climate-smart agriculture participation and climate change adaptation response on technical efficiency of rice production in three selected provinces of the Mekong Valley using the Propensity score matching approach. The cross-sectional data obtained through a face to face interview using a structured questionnaire consisting of 361 households recruited by administration selection.

The paper is quite well written and lucidly presented. The analysis is basically correct, and findings elaborately discussed.

Specific comments:

1.       In abstract, minor typographical error and missing words.

2.       The rationale and the well-connected literature to global situations added value to the paper

3.       In the methodological framework, writers rightly noted that the PSM approach has three analytical steps which were appropriately considered in the discussion section. However, there is a need to elaborate on the SFA procedure to determine technical efficiency in the methodology section. This is a key stage and will provide a better understanding of the PSM analysis

4.       In the study site section, a diagrammatic representation in the form of a map of the study area may add quality to the article.

5.       In the data description section, the reason for the exclusion of the third region (Tra Vinh) in Tables 1, 2, and 3 should be clearly stated.

6.       In the concluding section, the policy implications of the paper were briefly and generally highlighted, yet not fully addressed. There is a need to elaborate more on specific policy implications and particularly, implications for the individual provinces/regions under study.

Author Response

Response to Reviewer 3 Comments

Comments and Suggestions for Authors

The paper aims to assess the effects of climate-smart agriculture participation and climate change adaptation response on technical efficiency of rice production in three selected provinces of the Mekong Valley using the Propensity score matching approach. The cross-sectional data obtained through a face to face interview using a structured questionnaire consisting of 361 households recruited by administration selection.

The paper is quite well written and lucidly presented. The analysis is basically correct, and findings elaborately discussed.

Response:

We would like to thank you very much for your valuable comments. We have indicated all required changes in red color, all grammar check highlighted in yellow color, and all the other changes highlighted in grey color in the revised manuscript. We have revised the manuscript according to your comments and suggestions as follows.

1.     In abstract, minor typographical error and missing words.

Response 1:

We have deleted ‘of’ in the 15th line of page 1.

2.     The rationale and the well-connected literature to global situations added value to the paper

Response 2:

We would like to thank you so much for your positive comment

3.     In the methodological framework, writers rightly noted that the PSM approach has three analytical steps which were appropriately considered in the discussion section. However, there is a need to elaborate on the SFA procedure to determine technical efficiency in the methodology section. This is a key stage and will provide a better understanding of the PSM analysis

Response 3:

We have modified the sentence in 337th-339th line of page 10 as follows:

“Technical efficiency of rice farming in the Vietnamese Mekong Delta was parametrically estimated to be 0.7725 through the stochastic frontier analysis (SFA) approach (Table 4) (see also in [42]).”

à“Ho and Shimada [44] investigated the technical efficiency of rice farming in the Vietnamese Mekong Delta to be 0.7725 through a parametric approach of stochastic frontier analysis (SFA) (Table 4).”

In addition, SFA approach was used in another paper to determine the technical efficiency of rice farming. And we used this technical efficiency as an outcome variable for estimating the treatment effects in the PSM analysis. Therefore, we would like to only provide the PSM approach and skip the description of SFA procedure in this manuscript.

4.     In the study site section, a diagrammatic representation in the form of a map of the study area may add quality to the article.

Response 4:

According to your helpful suggestion, we have added a map of the study areas (namely Figure 1. Map of the study areas) to the manuscript.

5.      In the data description section, the reason for the exclusion of the third region (Tra Vinh) in Tables 1, 2, and 3 should be clearly stated.

Response 5:

Since the variable of Region is a category variable (including three categories of Long An, Ben Tre and Tra Vinh province), the two dummy variables (Region_LongAn and Region_BenTre) are used in the Table 1, 2, 3 with a reference or comparison category of Region_TraVinh.

Similarly, the variable of Access to water sources with three categories of near, medium, and far. Therefore, the two dummy variables (Access to water source_Near and Access to water source_medium) are used in the Table 1, 2, 3 with a reference or comparison category of Access to water sources_Far.

We have added a note below the Table 1 as follows:

“Note: a Access to water sources (Far) represents a reference or comparison category. b Region (Tra Vinh) represents a reference or comparison category.”

6.     In the concluding section, the policy implications of the paper were briefly and generally highlighted, yet not fully addressed. There is a need to elaborate more on specific policy implications and particularly, implications for the individual provinces/regions under study.

Response 6:

The policy implications of incentives to strongly support adaptation response by providing precise and timely information associated with climate change adaptation practices and CSA suitable for specifically geographical location through agricultural extension services at local level in the study were suggested mainly based on the significant variables of the propensity score models (variables of extension, belief in climate change, and farm characteristics such as farm area and access to water sources).

Even though there are a substantial difference in adaptation responses among provinces based on the surveyed data described in the 329th-325th line of page 10, the variables of geographical location at the provincial level have no significance in the propensity score models. Therefore, we could not elaborate on specific policy implications for specific province in this study.

Agriculture EISSN 2077-0472 Published by MDPI AG, Basel, Switzerland RSS E-Mail Table of Contents Alert
Back to Top